# Antioxidant and Cytotoxic Activities of *Usnea barbata* (L.) F.H. Wigg. Dry Extracts in Different Solvents

**DOI:** 10.3390/plants10050909

**Published:** 2021-05-01

**Authors:** Violeta Popovici, Laura Bucur, Antoanela Popescu, Verginica Schröder, Teodor Costache, Dan Rambu, Iulia Elena Cucolea, Cerasela Elena Gîrd, Aureliana Caraiane, Daniela Gherghel, Gabriela Vochita, Victoria Badea

**Affiliations:** 1Department of Microbiology and Immunology, Faculty of Dental Medicine, Ovidius University of Constanta, 7 Ilarie Voronca Street, 900684 Constanta, Romania; violeta.popovici@365.univ-ovidius.ro (V.P.); victoria.badea@365.univ-ovidius.ro (V.B.); 2Department of Pharmacognosy, Faculty of Pharmacy, Ovidius University of Constanta, 6 Capitan Al. Serbanescu Street, 900001 Constanta, Romania; antoanela.popescu@365.univ-ovidius.ro; 3Department of Cellular and Molecular Biology, Faculty of Pharmacy, Ovidius University of Constanta, 6 Capitan Al. Serbanescu Street, 900001 Constanta, Romania; verginica.schroder@univ-ovidius.ro; 4Research Center for Instrumental Analysis SCIENT, 1E Petre Ispirescu Street, 077167 Ilfov, Romania; teodor.costache@scient.ro (T.C.); dan.rambu@scient.ro (D.R.); iulia.cucolea@scient.ro (I.E.C.); 5Department of Pharmacognosy, Phytochemistry, and Phytotherapy, Faculty of Pharmacy, Carol Davila University of Medicine and Pharmacy, 6 Traian Vuia Street, 020956 Bucharest, Romania; cerasela.gird@umfcd.ro; 6Department of Oral Rehabilitation, Faculty of Dental Medicine, Ovidius University of Constanta, 7 Ilarie Voronca Street, 900684 Constanta, Romania; aureliana.caraiane@365.univ-ovidius.ro; 7Institute of Biological Research Iasi, Branch of NIRDBS, 47 Lascar Catargi Street, 700107 Iasi, Romania; daniela.gherghel@icbiasi.ro (D.G.); gabriela.vochita@icbiasi.ro (G.V.)

**Keywords:** *U. barbata*, usnic acid, UHPLC, polyphenols, tannins, polysaccharides

## Abstract

Lichens represent a significant source of antioxidants due to numerous metabolites that can reduce free radicals. *Usnea barbata* (L.) F.H. Wigg. has been recognized and used since ancient times for its therapeutic effects, some of which are based on its antioxidant properties. The present study aims to analyze the phytochemical profile and to evaluate the antioxidant and cytotoxic potential of this lichen species. Five dry extracts of *U. barbata* (UBDE) in different solvents (acetone, ethyl acetate, ethanol, methanol, water) were prepared by refluxing at Soxhlet to achieve these proposed objectives and to identify which solvent is the most effective for the extraction. The usnic acid content (UAC) was quantified by ultra-high performance liquid chromatography (UHPLC). The total polyphenols content (TPC) and tannins content (TC) were evaluated by spectrophotometry, and the total polysaccharides (PSC) were extracted by a gravimetric method. The 2,2-diphenyl-1-picryl-hydrazyl-hydrate (DPPH) free radical method was used to assess the antioxidant activity (AA) and the Brine Shrimp Lethality (BSL) assay was the biotest for cytotoxic activity evaluation. The ethyl acetate extract had the highest usnic acid content, and acetone extract had the highest content of total polyphenols and tannins. The most significant antioxidant effect was reported to methanol extract, and all the extracts proved high cytotoxicity. The water extract has the lowest cytotoxicity because usnic acid is slightly soluble in this solvent, and it was not found at UHPLC analysis. All extracts recorded a moderate correlation between the content of usnic acid, polyphenols, tannins, and AA; furthermore, it has been observed that the cytotoxicity varies inversely with the antioxidant effect.

## 1. Introduction

In the confrontation with illness, the human body is far from well-protected. Once installed at the molecular level, an imbalance [1] can sooner or later generate disease, and the human body could be hardly recovered. Not only that: it is increasingly challenging to avoid internal or external factors that expose the body to these imbalances, which generate the well-known oxidative stress [2]. Environmental pollution [3], heavy metals, xenobiotics, ultraviolet light [4], immune cell activation, inflammation [5], and mental stress [6] lead to excessive reactive oxygen and nitrogen species (ROS and RNS) [7]. They generate oxidative destruction of cellular macromolecules (nucleic acids, proteins, carbohydrates, lipids). The body antioxidant defense is no longer coping, and these reactive species negatively affect the cell structures, producing complex changes that generate premature ageing [8] and numerous diseases [9]. Current research in the medical world is focused on finding new organic compounds with an antioxidant role, protecting the human body against free radicals [10].

It is known that ROS and RNS are permanently generated in plants as a result of aerobic metabolism; some of these reactive species have high toxicity, and the numerous cellular mechanisms rapidly neutralize them [11]. Various plant extracts [12] and isolated natural compounds [13] have antioxidant effects: vitamins [14], flavonoids [15], polyphenols [16], sterols [17], and polysaccharides [18]. Lichens are a significant source of antioxidants [19]; the *Parmeliaceae* family [20] with the *Usnea* genus, are known due to their antioxidant metabolites [21]. The present study is performed on *U. barbata* (commonly called Old Man’s Beard, and also named Song Luo in China), a lichen used for over 2000 years in Chinese traditional medicine for its therapeutic properties. Its phytochemical profile consists of primary metabolites (fatty acids and lichen polysaccharides as lichenan, homoglucan) and specific secondary metabolites. This last category, named lichen secondary metabolites, consists of specific phenolic compounds (depsides, depsidones) [22] (p. 603), dibenzofurans (usnic acid), and diphenyl-ethers [23]. These organic constituents are the results of the special structure of lichens, represented by the symbiosis between fungus and algae (cyanobacteria) [24]. Usnic acid is the most significant secondary metabolite of the genus *Usnea,* with various biological activities: antioxidant, gastroprotective, cytoprotective, immunostimulatory, antimicrobial, anti-inflammatory, and antitumor [25]. The aim of this study is to perform a comparative analysis of five UBDE obtained using different solvents. We highlighted their phytochemical profile determining the content of usnic acid [26], polyphenols [27], tannins [28], and polysaccharides [29]—all these classes of organic compounds being recognized due to their antioxidant properties. The antioxidant activity of UBDE was evaluated and correlated with UAC, TPC, and TC.

In other circumstances, plants appear to generate ROS as signaling molecules to control various processes, including pathogen defense and programmed cell death [11]. For this reason, the last part of our study consists of analyzing the cytotoxic action of all five extracts; we also included in this study usnic acid because it has a dual role in redox processes [30]. The effects of UBDE were evaluated using *Artemia* sp. (also named brine shrimp) bio-tester organisms, BSL assay [31], used for the pre-screening of plant extracts cytotoxicity [32]. The BSL assay represents a significant step before modelling in pharmaceutical research [33]. Our previous studies on UBDE in acetone proved in vivo cytotoxic effects on *Artemia* sp. larvae and in vitro antitumor activity on human tongue squamous cells carcinoma (CAL 27 cell line) by involving the apoptotic mechanism [34]. *U. barbata* dry extract in acetone has a high UAC, and usnic acid induces ROS-dependent apoptosis of tumor cells [35,36].

The biological activity of different species of lichens is a consequence of the natural mixtures of compounds and, also of their interactions. This study analyzed the variable results obtained for each extract, correlating them with the chemical constituents extracted by each solvent separately. An overview of the antioxidant and cytotoxic potential variation in *U. barbata* dry extracts was suggested at the end of this study.

## 2. Results

### 2.1. Preparation of Usnea barbata (L.) F.H. Wigg. Dry Extracts and Determination of Metabolites Content

#### 2.1.1. Lichen Extraction Yield

*U. barbata,* freshly harvested and dried, had a grey-green color, a fresh smell, and a spicy taste. The obtained value of the loss on drying was 10.94 ± 0.94% for the dried lichen.

The dry extracts of *U. barbata* prepared in five different solvents had various colors, depending on the other organic compounds extracted in each solvent. The obtained yields are reported in Table 1.

It can be noted that UBDE in water revealed the lowest yield; the extracts in acetone and ethyl acetate showed approximately three times higher products than in water, and their values are very similar (6.36% for acetone extract and 6.27% for ethyl acetate extract). The highest yield was on UBDE in ethanol (12.52%) and methanol (11.29%), about two times higher than in both previous solvents. The color and the physical properties of all five UBDE were different, varying from yellow-brown (acetone extract) to dark brown-reddish (water extract) (Table 1).

#### 2.1.2. UHPLC Determination of the Usnic Acid Content

The corresponding chromatograms were represented in Figure 1. Other peaks can be recorded in the following chromatograms, corresponding to other metabolites extracted in UBDE in methanol, ethanol, and water. Figure 1d–f shows that the RT values for these organic compounds were lower than RT for usnic acid (2 min < RT < 3 min).

All of these UHPLC determination data are resumed in Table 2, in which the UAC was reported as mg of usnic acid per gram UBDE. The obtained results illustrated in Table 2 showed that usnic acid was quantified in the highest content in UBDE in ethyl acetate (376.73 mg/g), followed, in decreasing order, by UBDE in acetone (282.78 mg/g), in methanol (137.60 mg/g) and ethanol (127.21 mg/g); UBDE in water does not contain any usnic acid.

This UHPLC-PDA quantitative determination method was validated for the reference substance, usnic acid; it was the subject of another previously published paper [37]. The linearity was verified by the least-squares procedure, on the 125–2500 mg/g range (sample units), for a value of R^2^ of 0.99988. The accuracy expressed as percentage relative error is 2.26%. The precision, calculated as repeatability at the concentration 1250 mg/g, and presented as relative standard deviation (RSD), is 1.16%. The limit of quantification (LOQ) was determined at 2.5 mg/g with a “signal-to-noise” ratio = 14:1. The limit of detection (LOD) was 1.25 mg/g. Peak purity index was assessed by spectral reprocessing using Chromera software, on 240–700 nm range at 15% of peak height. Visual representation (contour map) of the full spectrum plotted on time was obtained in the same manner on 190–700 nm; moreover, peak identity was confirmed by matching analyte peak spectra and retention times extracted from chromatogram with reference to substance ones. All these data were detailed in the Appendix A.

#### 2.1.3. Determination of the Total Polyphenols Content

The total polyphenol content is expressed in mg equivalents of pyrogallol per g UBDE (mg PyE/g UBDE). The obtained data analysis revealed that UBDE in acetone had the highest TPC: 101.09 ± 0.5 mg PyE/g UBDE. The lowest TPC values (very closed) were achieved in UBDE in ethyl acetate (42.40 ± 1.4 mg PyE/g UBDE) and water (45.8 ± 1.2 mg PyE/g UBDE). The UBDE in both alcohols showed similar TPC values: 67.3 ± 0.5 mg PyE/g UBDE in ethanol and 70.7 ± 1.7 mg PyE/g UBDE in methanol.

The data obtained in TPC determination were registered in Table 2.

#### 2.1.4. Determination of the Tannins Content

The highest content in TC was found in UBDE in acetone (24.4 ± 0.6 mg PyE/g UBDE), followed, in decreasing order, by UBDE in ethanol (14.7 ± 0.05 mg PyE/g UBDE), in methanol (9.99 ± 1.7 mg PyE/g UBDE) and ethyl acetate (3.85 ± 0.26 mg PyE/g UBDE). UBDE in water revealed the lowest TC value (1.31 ± 0.2 mg PyE/g UBDE), as seen from Table 2. Based on the data from Table 2, Figure 2 could offer a relevant comparative image of the content of the secondary metabolites in all U. barbata dry extracts.

#### 2.1.5. Determination of the Total Polysaccharides Content

From 50.173 g dried lichen, 2.704 g polysaccharides were obtained; subtracting the value of the loss on drying of lichen, the yield of this process was 5.39%.

### 2.2. Evaluation of the Antioxidant Activity

The obtained results analysis could note that UBDE in methanol had the highest antioxidant activity (DPPH IC_50_ = 3300 µg/mL). The following UBDE, in ethanol and acetone, had very similar values of DPPH IC_50_ (4462 µg/mL and, respectively, 4608 µg/mL). UBDE in water had DPPH IC_50_ = 6211 µg/mL, the lowest antioxidant activity was registered by UBDE in ethyl acetate, with DPPH IC_50_ = 7701 µg/mL (Table 3).

Each type of UBDE has revealed a directly proportional relationship between the concentration of the extracts and their antioxidant activity.

The correlation between TPC and AA expressed by % scavenger DPPH could be evaluated by linear trendlines, correlation coefficients (R^2^), and linear equations for each tested extract. The R^2^ values showed a high correlation for UBDE in methanol (R^2^ = 0.9453) and ethanol (R^2^ = 0.9308), a moderate correlation for UBDE in water (R^2^ = 0.7250) and acetone (R^2^ = 0.7051), and a low correlation for UBDE in ethyl acetate (R^2^ = 0.5408).

The same model could evaluate the correlation between TC and AA; the linear equations and R^2^ values are registered in Table 4.

The correlations between UAC, TPC, TC (mg/g UBDE), and AA (expressed by DPPH IC_50_ mg/mL) of all UBDE were investigated by using regression analysis (Figure 3a–c).

Analyzing the linear trendlines, linear equations, and regression coefficients R^2^ reported in Figure 3 it could be noted that there exists a moderate correlation between DPPH IC_50_ and UAC (R^2^ = 0.6374) and TC (R^2^ = 0.5672) and a low correlation between DPPH IC_50_ and TPC (R^2^ = 0.3525).

The antioxidant activity of the tested UBDE (expressed by DPPH IC_50_) in correlation with their phytochemical profile was shown in Figure 4a.

Figure 4b highlights the correlation between the phytochemical profile of UBDE and their cytotoxic activityIt can be seen that the cytotoxicity of UBDE varies directly proportional to the content of secondary metabolites. Thus, UBDE in ethyl acetate and acetone, with the greatest UAC, have the highest levels of cytotoxicity (Figure 4b) and, at the same time, a low antioxidant effect (Figure 4a).

### 2.3. Evaluation of the Cytotoxic Activity by Brine Shrimp Lethality Assay

Six stock solutions in DMSO (mg/mL) of usnic acid and UBDE were obtained; three different usnic acid and UBDE dilutions in DMSO 0.1% were tested (Table 5). All the UBDE and usnic acid reported cytotoxic effects directly proportional to the concentrations of the tested solutions.

Evaluation of the cytotoxicity of the UBDE showed the appearance of effects for all extracts, except for UBDE in ethyl acetate (32.4 µg/mL) and UBDE in water (32, 160, 480, 320 µg/mL) (Table 5). At concentrations above 100 µg/mL (1:10), the recorded effects indicate low toxicity (mortality between 20% and 38%).

The brine shrimp larvae were exposed to concentrations over 300 µg/mL; they were affected by the content of the tested extract, and the registered mortality was significant, except UBDE in water (Table 5).

Comparing the results between the effects induced by UBDE in different solvents and the usnic acid, reported to the negative control (water) by Dunnett test, the statistical analysis showed significant differences (*p*-value < 0.05) in the case of exposed lots (Table 6). The mortalities were statistically significantly different between negative control samples and treated groups. The diversity of interactions induced by the composition of the extracts that possibly influence the statistical response significance, below the statistical significance level, is the 1:10 fold dilution group (Table 6).

Analyzing the data registered in Table 6; Table 7, through the Meyer [38] and Clarkson’s [39] toxicity index it could be mentioned that UBDE in water had non-cytotoxic activity (BSL LC_50_ = 1983.68 µg/mL), followed, in increasing order, by usnic acid (BSL LC_50_ = 424.75 µg/mL) and acetone (BSL LC_50_ = 411.77 µg/mL), ethanol (BSL LC_50_ = 338.39 µg/mL) and methanol (BSL LC_50_ = 250.19 µg/mL). The highest cytotoxic activity was recorded by ethyl acetate extract (BSL LC_50_ = 219.59 µg/mL) (Table 7). All the results obtained in this analysis were calculated using the Probit method, and the lethality (LC_50_ and LC_100_ and 95% interval confidence) was registered in Table 7.

Microscopic observations showed general morphological changes (Figure 5), such as body deformation (Figure 5c–e) and shedding disturbance (Figure 5c,d). Inhibition of larval development was also observed compared to unexposed organisms (Figure 5a). Evident at the cellular level is the accumulation of cytoplasmic inclusions and the loss of intercellular connections. These cytological phenomena are significant in UBDE in methanol, ethyl acetate, and acetone (Figure 5c–e).

## 3. Discussion

The secondary metabolites and the lichen biological activities are influenced by environmental factors such as seasonal variation, temperature, light, and habitat. The correlation between the lichen habitat and their usnic acid content was shown by Cansaran et al. (2008); the lichens have the highest usnic acid content at the altitudes between 700 and 1500 m because the water remains liquid—the probability of water remaining as ice is higher above this elevation [40]. Loss on drying was calculated for the lichen sample to know the accurate weight of dried lichen, subtracting loss on drying value from the weighed mass.

For obtaining *U. barbata* extracts, the following five solvents: acetone, ethyl acetate, ethanol, methanol, and water, were used. They are most frequently used for the preparation of plant extracts, solubilizing most phytoconstituents. Determining TPC and AA of various plant extracts in water, ethanol, and acetone, Dirar et al. (2019) [41] (p. 263) provided that acetone extracts had the highest TPC for the numerous studied species. Comparing the ethanol and methanol plant extracts, Sultana et al. (2009) [42] showed that methanol extracts had the highest TPC. In our study, TPC values decreased in order: UBDE acetone, methanol, and ethanol. Polysaccharides are, generally, soluble in water [43]; the solubility of usnic acid increases in order: water, ethanol, methanol, acetone, and ethyl acetate [44]. Refluxing at Soxhlet 8 h, followed by filtration and concentration of the obtained extract, represent one of the most efficient extraction methods. From 400 g of *U. longissima* dried lichen, Maulidiyah et al. (2011) [45] obtained 28.79 g dry acetone extract; the yield of the extract was slightly higher (7.19%) than that obtained in our present study (6.36%). The total polyphenol content was determined in various extracts of *Usnea* sp.: *U. florida* (methanol extract: 10.5 mg/g, water extract: 10.4 mg/g); *U. gattensis* (acetone extract: 14 mg/g, methanol extract: 35 mg/g); *U. longissima* (methanol extract: 38.6 mg/g, water extract: 18.3 mg/g) [46].

The variable content of usnic acid in various *U. barbata* extracts depended on the used solvents and extraction methods. Zugic et al. (2016) analyzed the usnic acid content in four *U. barbata* different extracts: supercritical CO_2_ extract, Soxhlet extracts (ether and ethanol fractions), and 70% ethanol macerate; the obtained values decreased in order: 81.41% usnic acid in supercritical CO_2_ extract, 67.09% in ether fraction, 2.43% in ethanol fraction and 1.39% in 70% ethanol macerate [47].

Analyzing the obtained results of the antioxidant activity, we could observe that UBDE in methanol and ethanol presented a higher AA than UBDE in acetone even if their TPC was lower than in it. This result can be due to the polyphenols soluble in alcohols with a higher antioxidant effect than other specific phenolic compounds extracted in acetone. In their study, Dirar et al. (2019) [41] (pp. 263–265), describing similar results, stated that phenolic compounds with free hydroxyl groups have intense free radical scavenging activities. The high correlation between TPC and AA available in UBDE in methanol and ethanol could be explained similarly. The water extract contains polyphenols and polysaccharides [48], and both classes of organic compounds [49] have antioxidant activity [50]; UBDE in ethyl acetate, which also had the lowest TPC, reported the lowest AA. It is essential to report the high correlation between AA of all UBDE (% scavenger DPPH) and their TPC (Table 4).

In their studies on *U. barbata* acetone extract, Rancovic et al. (2012) [51] found a TPC value of 31.3 µg/mL, DPPH IC_50_ = 667.9 µg/mL, and a high correlation between TPC and AA. In the scientific literature, numerous studies evaluate the antioxidant potential of lichens from various Earth zones by different methods [52]. It has been shown that the lichen extracts in different solvents and their isolated metabolites have an antioxidant activity [53]. The BSL assay cannot determine the mechanism of action of the metabolites from the tested *U. barbata* extracts; it can only provide a preliminary screening that can be followed by more specific bioassays. Lethality and cytological changes in brine shrimp larvae are easy to assess in these organisms [54], making screening very fast and efficient. A study on liver cells [55] showed an increase in lipid droplet content and fragmentation of the endoplasmic reticulum in conditions of exposure to usnic acid. We consider these cytological aspects similar to those observed in the organisms exposed in UBDE (Figure 5b–e).

Many researchers evaluated the biological potential of the lichen extracts and isolated constituents; also, usnic acid is recognized as the most specific and bioactive lichen secondary metabolite. Synthesizing the obtained results, the influence of UAC from UBDE on the studied biological activities could be analyzed. Significantly, UBDE in ethyl acetate, which has the highest UAC, has the lowest antioxidant effect, and also the highest cytotoxicity. Numerous studies highlighted both biological effects of the various lichen extracts and usnic acid. Generally, the antioxidant capacity and cytotoxicity are considered opposite activities concerning living cells. Thus, it could be expected that a compound (or plant extract) with a high antioxidant effect to achieve high cellular protection and to have low cytotoxic action. It was observed that the *U. barbata* dry extracts with high cytotoxic effects show reduced levels of antioxidant activity.

Analyzing the studied UBDE, it can be observed that the five solvents selected for extraction are frequently used in medical and pharmaceutical research laboratories. In the Pfizer solvent selection guide (according to the Green Chemistry concept) [56] (pp. 3–6), these solvents are included in the “preferable” category, with the lowest toxicity and the highest safety. Acetone, ethanol and methanol are miscible with water; only ethyl acetate has shown a low solubility in water [56] (p. 4). This study suggests acetone, ethyl acetate, ethanol, and methanol as suitable solvents for *U. barbata* extracts, indicating the proving arguments. Thus, both alcohols provided the highest extraction yields of UBDE; this advantage is accompanied by a wide range of biologically active compounds. The extraction yields in acetone and ethyl acetate are about two times lower; however, both UBDE have a significant UAC because usnic acid presents optimal solubility in these two solvents. *U. barbata* dry acetone extract has a substantial UAC. It also contains other secondary metabolites with various biological effects. Instead, if the main objective is to obtain UBDE with the highest content of usnic acid, ethyl acetate would be the most appropriate solvent.

## 4. Materials and Methods

### 4.1. Lichen Extraction Yield

For the present studies, *U. barbata* was harvested from a region located at 900 m altitude from the Călimani mountains (Suceava county, Romania) in March 2020 because the lichen secondary metabolites are at maximum level [57] in early winter or early spring and a minimum level during the summer [58].

*U. barbata* was manually harvested directly from the branches of conifers. The fresh lichen was cleaned of impurities and dried at 18–25 °C, in a herbal room, sheltered from the sun rays. After drying, the obtained herbal product was preserved for a long time in the same conditions for use in subsequent studies. The lichen species identification was performed by the Department of Pharmaceutical Botany of the Faculty of Pharmacy, Ovidius University of Constanta, using standard methods.

A weighing ampoule brought to constant weighed together with the lichen sample was kept in the oven at 105 °C, for two hours, and then cooled in the desiccator and weighed. The drying process continued in the oven for one hour, followed by cooling and weighing, until the constant weight was achieved [59].

The dried lichen was ground to a powder and extracted for eight hours with each solvent (acetone, ethyl acetate, ethanol, methanol, water) in a Soxhlet continuous reflux system. Extraction was different for each extract, being around the boiling point of each solvent. After filtration, the water extract was concentrated on rotavapor Butchi R-215 with a vacuum controller V-850 lyophilized a with freeze-dryer Christ Alpha 1–2 B Braun Biotech International with vacuum pump RZ 2.5.

In the other four *U. barbata* extracts, the rotary evaporator TURBOVAP 500 Caliper was used for evaporation of the solvents. Next, these extracts were kept for 16 h in a chemical exhaust hood for each optimal solvent evaporation.

The obtained dry extracts were transferred to sealed-glass bottles and stored in the freezer (Sirge FREEZER) at −24 °C until processing.

### 4.2. UHPLC Determination of the Usnic Acid Content

Usnic acid was separated in UBDE dissolved in DMSO following a chromatographic column filled with reverse stationary phase type C18. After elution from the column, the compounds were analyzed using the Photodiode Array (PDA) Detector; the signal corresponding to the target compound was recorded at a wavelength of 282 nm [37].

The PerkinElmer^®^ Flexar^®^ FX-15 UHPLC system fitted with a Flexar FX PDA-Plus photodiode array detector was the platform for this analysis (UHPLC-PDA). The Brownlee Analytical C18 column is filled with 5 µm superficially porous particles; it has an inner diameter of 4.6 mm and a length of 150 mm [60].

Working conditions consisted of: flow = 1.5 mL/min; temperature in the column compartment = 25 °C; injection volume = 20 µL; analysis time: 10 min. The mobile phase was an isocratic system methanol/water/glacial acetic acid (80:15:5). The samples were UBDE dissolved in acetone, ethyl acetate, ethanol, methanol, and water and diluted to 1:50 with DMSO. The standard-stock solution was prepared by dissolving 20 mg of usnic acid (Sigma-Aldrich, St. Louis, MO, USA) in 10 mL DMSO.

The standard solutions (Scal) were prepared from the standard-stock solution; the following concentrations were obtained: 10, 20, 50, 100, and 200 µg/mL, with which the calibration curve was drawn: (y = (48.46290 × 10^3^)x + (−40.16791 × 10^3^); R^2^ = 0.99988) (Figure 6).

The Quality Control (QC) solutions were prepared by adding 20 µg standard-stock solution in a volumetric flask of 10 mL and completing with DMSO up to the mark.

Two samples of QC solutions of 40 µg/mL were injected at the beginning and at the end of the sequence to assure the accuracy of the analysis. Accuracy between 97.7–98.8 indicates that the analysis is highly accurate.

The conversion of the standard solution concentration into the sample concentration was calculated using the formula:C_smpl.UA_ = C_std.UA_/1000 ∗ D/5 ∗ 100
where: C_smpl.UA_ is the usnic acid concentration of the sample; C_std.UA_ is the usnic acid concentration of Standard Solution; D is the sample dilution factor—according to this procedure, D value = 50. The verification of the method accuracy was realized by comparing the QC standard theoretical concentration with the concentration obtained from its analysis. This formula was used to calculate the accuracy of the method:AQC% = CcQC/CTQC ∗ 100

AQC% is the accuracy of determining the QC solution; CcQC is the concentration of the injected QC solution; CTQC is the theoretical concentration of the QC solution. The chromatogram of standard solution 200 µg/mL (Scal-1) shows that the usnic acid retention time (RT) was about 3.640 min (Section 2.1.2, Figure 1).

All the results were obtained using PerkinElmer Chromera Manager Software, on HP ProDesk 400 G1 MT Intel^®^ Core™ i5-4570 PC.

### 4.3. Determination of the Total Polyphenols Content

The total polyphenols content was determined with Folin-Ciocâlteu reagent (phosphomolybdotungstic acid) using a method provided by Maisetta et al. [61] Pyrogallol was used as standard, and the TPC values were calculated as mg of Pyrogallol equivalents (PyE) per g UBDE.

For this analysis, to five volumetric flasks of 25 mL, 5 mL of each UBDE (A1-A5) was added, filtered through 99 filter paper. They were completed up to the sign with the same solvent, and B1-B5 solutions were obtained. In five volumetric flasks of 25 mL, 2 mL of each solution B1-B5 were added, with 1 mL of Folin-Ciocâlteu reagent, 10 mL water, and 290 g/L of Na_2_CO_3_ solution, up to the mark; a blue coloration resulted in each volumetric flask. After 30 min of reaction at room temperature [62], the absorbencies (each value was noted with A1 in the calculation formula) were determined at 760 nm, using a Jasco V630 UV-Vis Spectrophotometer (Japan) with Spectra Manager™ Software.

All the determinations were run in triplicate; using the Microsoft Excel software (Microsoft Corporation, Redmond, WA, USA), the standard deviations (SD) and the mean values were calculated.

### 4.4. Determination of the Tannins Content

According to a previous study reported by Galvao et al. (2018) [63], the tannins content was determined. The procedure consists of three phases: determination of TPC in different UBDE extracts by Folin-Ciocâlteu method (Section 4.3.) absorption of tannins on standardized hide-powder, and determination of the phenolic compounds in the solution remaining after the second phase. The quantification of the molybdenum oxides blue coloration intensity was determined by spectrophotometry (760 nm). The difference between both determinations even represents the tannin content. All the determinations were performed in triplicate; the data were reported as means ± SD using the Microsoft Excel software (Microsoft Corporation, Redmond, WA, USA).

### 4.5. Determination of the Polysaccharides Content

Total polysaccharides were extracted using a classical gravimetric method with ethanol precipitation described in a recent study by Tikhomirova et al. (2020) [64] Approximately 50 g of dried, chopped lichen was refluxed for 30 min with 500 mL of distilled water. The extract was filtered by filter paper in a 250 mL volumetric flask and completed with water up to the mark. The entire volume of aqueous extract was added in a thin stream on 2000 mL of 96% ethanol under continuous stirring. The obtained mucilage precipitate was separated from the liquid phase by filtration through filter paper; then, it was dried in the oven at 105 °C for 3 h, until constant weight. It was kept in the desiccator until it cooled to room temperature, and then it was weighed.

### 4.6. Evaluation of the Antioxidant Activity

The antioxidant activity was determined on a Jasco V630 UV-Vis Spectrophotometer (Japan) using a DPPH free radical scavenging assay [65].

The DPPH solution was prepared by dissolution of DPPH (Sigma Aldrich) in methanol to obtain an absorbance value of 0.8 ± 0.02; 0.1 of each UBDE was vortexed with 3.9 mL of DPPH solution for 30 s. The reaction time at room temperature was 30 min; finally, the absorbance at 515 nm was recorded. The DPPH solution with no added extract was used as control, and methanol was used as a blank. Usnic acid was dissolved in acetone to obtain a solution with a similar concentration as the lichen solutions. The concentration of the usnic acid solution was 0.2 mg/mL. For each UBDE, the following dilutions were obtained: 1:1, 1:2, 1:3, 1:4, 1:5. The scavenger activity was calculated as follows:Scavenging of DPPH (%) = 100 × [(A control − A sample)/A control]

A control and A sample being the absorbance values at 515 nm for DPPH solution and UBDE solution.

All the determinations were completed in triplicate; the obtained data were registered as means ± SD and analyzed using linear regression analysis with the Microsoft Excel software (Microsoft Corporation, Redmond, WA, USA).

### 4.7. Evaluation of the Cytotoxic Activity by Brine Shrimp Lethality Assay

Brine shrimp larvae were obtained by introducing the cysts of *Artemia salina* L for 24 h, in a saline solution of 35%, under conditions of continuous light and aeration. After hatching brine shrimp in the first larval stage (instar I), they were separated and introduced into experimental pots (with a volume of 1 mL) in 2–3% saline solutions [66]. For these tests, six stock solutions of usnic acid and five different UBDEs were prepared by solubilization in DMSO 0.1%. *Artemia salina* L. larvae were not fed during the test period to not interfere with the tested extracts. This bioassay was valid for 24–48 h, during which the larvae had embryonic energy reserves. Brine shrimp larvae were exposed to different usnic acid concentrations and various UBDE; they were evaluated periodically, recording the antennae movements and the larvae metamorphosis from the first stage to the second and third stages [67].

After 24 h of exposure, the death rate was the measurable parameter for quantifying larvae response to the various concentrations of usnic acid and UBDE. For control, 3% saline solution and 0.1% DMSO in saline solution were used to evaluate solvents effect on *Artemia salina* L. For each concentration of usnic acid and UBDE, four repetitions were performed.

The statistical analysis of biological data, and the larvae mortality, was calculated as a mean for four repetitions ± SD. One-way ANOVA, by Dunnett test, was used for evaluated experimental groups vs. negative control. Statistically significant differences were considered for *p*-values < 0.05.

## 5. Conclusions

All *Usnea barbata* (L.) F.H. Wigg. extracts highlight antioxidant activity due to their secondary metabolites (usnic acid, polyphenols, and tannins) by reducing the free radicals. The water extract showed no cytotoxic activity on brine shrimp larvae. However, all the other UBDEs obtained in our study and the usnic acid proved high cytotoxicity on *Artemia salina* L. larvae.

Our study novelty consists of the comparative analysis of five dry extracts of *U. barbata* with the usnic acid. These extracts were obtained in five safe solvents, and the solubility of usnic acid increases from water to ethyl acetate. The objectives were to evaluate the extraction yield, the content of active phytoconstituents and their biological activities. The analysis of two opposite biological effects—the antioxidant activity (considered as cytoprotective) and the cytotoxic action—for optimal correlation with the metabolites content of each extract has been described. The obtained data could enrich the existing information in the scientific database, which must be constantly updated by quantifying *U. barbata* secondary metabolites responsible for antioxidant and cytotoxic activity.

These presented results create the premise for further studies focused on highlighting and quantifying the in vitro antitumor properties and deciphering the mechanisms of action of various *U. barbata* dry extracts on different human cell lines.

## Figures and Tables

**Figure 1 plants-10-00909-f001:**
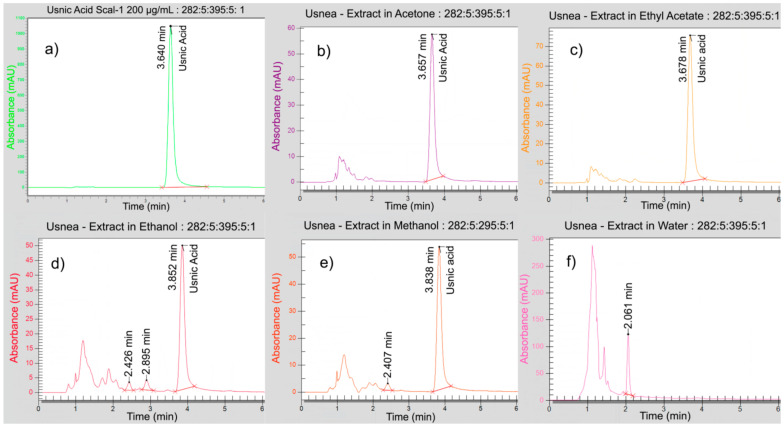
The chromatograms of usnic acid standard (**a**) and UBDE in different solvents (**b**–**f**); the baseline marked lines represent the peak integration.

**Figure 2 plants-10-00909-f002:**
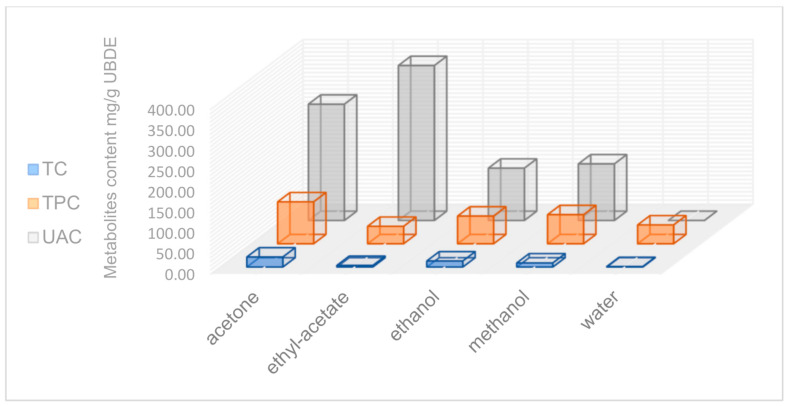
The secondary metabolite content (UAC, TPC, and TC) in UBDE in different solvents.

**Figure 3 plants-10-00909-f003:**
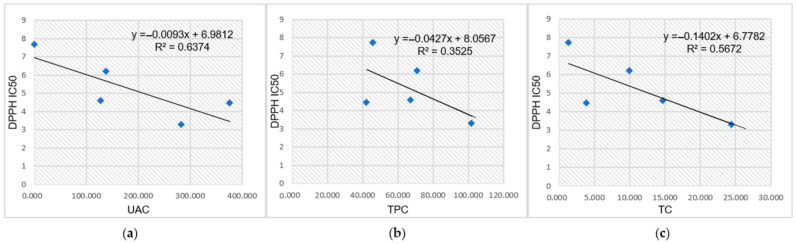
The correlation between UAC (**a**), TPC (**b**), TC (**c**), and AA (expressed by DPPH IC_50_) of UBDE.

**Figure 4 plants-10-00909-f004:**
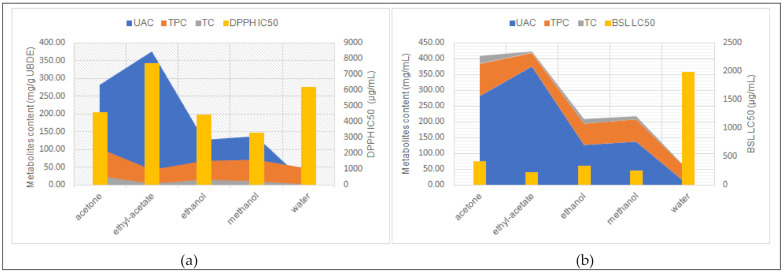
The correlation between the phytochemical profile of UBDE and antioxidant activity (**a**) and cytotoxic effect (**b**), expressed by DPPH IC_50_ and BSL LC_50_, respectively.

**Figure 5 plants-10-00909-f005:**
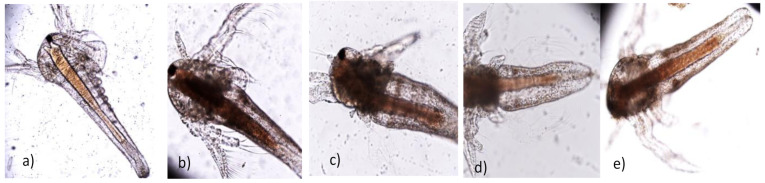
Microscopic details of exposed larvae—(**a**) negative control; (**b**) UBDE in water; (**c**) UBDE in methanol; (**d**) UBDE in ethyl acetate; (**e**) UBDE in acetone (magnification ×100).

**Figure 6 plants-10-00909-f006:**
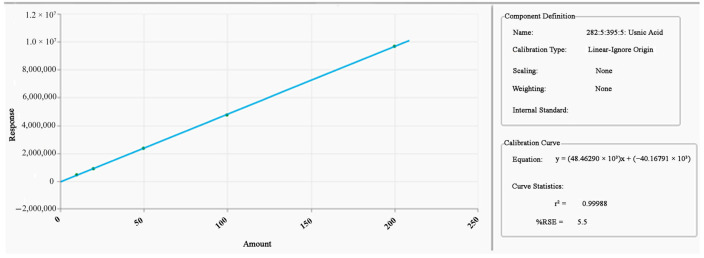
The calibration curve of usnic acid.

**Table 1 plants-10-00909-t001:** The temperature values for refluxing at Soxhlet, extraction yield, and colors of UBDE using different solvents.

UBDE	Temperature of Extraction	Yield (%)	UBDE Color
Acetone extract	55–60 °C	6.36	Yellow-brown
Ethyl acetate extract	75–80 °C	6.27	Brown-yellow
Ethanol extract	75–80 °C	12.52	Light brown
Methanol extract	65 °C	11.29	Brown
Water extract	95–100 °C	1.98	Dark brown-reddish

**Table 2 plants-10-00909-t002:** The metabolites content of UBDE in different solvents.

UBDE	UAC mg/g UBDE	TPC ± SD (mg PyE/g UBDE)	TC ± SD (mg PyE/g UBDE)
Acetone extract	282.78	101.09 ± 0.50	24.4 ± 0.60
Ethyl acetate extract	376.73	42.40 ± 1.40	3.85 ± 0.26
Ethanol extract	127.21	67.3 ± 0.50	14.7 ± 0.05
Methanol extract	137.60	70.7 ± 1.70	9.99 ± 1.70
Water extract	0.00	45.8 ± 1.20	1.31 ± 0.20

**Table 3 plants-10-00909-t003:** DPPH IC_50_ values for tested UBDE.

UBDE	Acetone Extract	Ethyl Acetate Extract	Ethanol Extract	Methanol Extract	Water Extract
DPPH IC50 (µg/mL)	4608	7701	4462	3300	6211

**Table 4 plants-10-00909-t004:** The correlation between TCP and TC values and AA (% scavenger DPPH) for each UBDE.

Parameter	UBDE	Acetone Extract	Ethyl acetate Extract	Ethanol Extract	Methanol Extract	Water Extract
TPC	Linear equation	y = 0.0527x + 5.189	y = 0.0873x + 0.9336	y = 0.0988x + 5.2527	y = 0.104x + 3.5547	y = 0.0861x + 4.0976
	R^2^ value	0.7051	0.5408	0.9308	0.9453	0.725
TC	Linear equation	y = 0.22x + 5.1893	y = 0.9576x + 3.9446	y = 0.4525x + 5.2502	y = 0.7356x + 3.5564	y = 2.9895x + 4.1226
	R^2^ value	0.705	0.5383	0.9309	0.9452	0.7221

**Table 5 plants-10-00909-t005:** The BSL assay protocols and larval mortality (%) mean values ± SD for an experimental four repetitions.

Stock Solutions (mg/mL)	Usnic Acid	UBDE in Acetone	UBDE in Ethyl acetate	UBDE in Ethanol	UBDE in Methanol	UBDE in Water		Water/DMSO (0.1%)
	12.9	17.2	16.2	16.1	16.1	16.0 ^dil^	16.0 *	Control Samples
**Tested samples**	**Concentrations (µg/mL)**							
1:50	24.8	34.4	32.4	32.2	32.2	32	320	-
1:10	160	172	162	161	161	160	1600	-
1:3.4	387	516	486	483	483	480	4800	-
**Mortality (%)**								
1:50	8.96 ± 7.60	14.94 ± 1.55	0.00	15.28 ± 3.69	23.51 ± 11.43	0	0	0
1:10	20.95 ± 6.32	38.82 ± 12.6	33.53 ± 14.7	29.37 ± 13.1	24.02 ± 3.35	0	50 ± 5.77	0
1:3.4	40.07 ± 17.8	54.24 ± 16.9	100 ±0	75.54 ± 28.2	87.82 ± 15.8	0	100 ± 0	0

^dil^ UBDE in water samples 1:50; 1:10; 1:3.4; * Additionally UBDE in water samples with 1:5; 1:1; 1:0.3 dilutions.

**Table 6 plants-10-00909-t006:** The analysis of variance (one way) for all tested samples; the analyzed groups were exposed to different concentrations.

ANOVA							
Source of Variation	d.f.	SS	MS	F	*p*-Value	F Crit	Omega Sqr.
1:50	6	2113.23	352.20	5.59	0.013	3.81	0.50
1:10	6	5932.10	988.68	1.76	0.16	3.81	0.14
1:3.4	6	32,172.06	5362.01	22.70	0.00000004	3.81	0.82

**Table 7 plants-10-00909-t007:** The BSL assay results (LC_50_ and LC_100_) and toxicity evaluation of usnic acid and UBDE tested in 24 h exposure.

Samples	LC_50_ (µg/mL)	Confidence Interval 95%		LC_100_ (µg/mL)	Toxicity Index *
		Lower	Upper		
Usnic acid	424.75	342.20	507.31	923.42	highly toxic
UBDE in acetone	411.77	274.26	549.29	1103.41	highly toxic
UBDE in ethyl acetate	219.59	176.09	263.09	359.70	highly toxic
UBDE in ethanol	338.39	265.02	411.74	808.53	highly toxic
UBDE in methanol	250.19	182.91	318.28	593.78	highly toxic
UBDE in water	1983.68	1455.84	2511.52	3428.64	non-toxic

* Toxicity index: LC_50_ > 1000 µg/mL non-toxic, between 500–1000 µg/mL low or moderate toxic and 100–500 µg/mL highly toxic.

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
