# Peer review of "Antioxidant and Cytotoxic Activities of Usnea barbata (L.) F.H. Wigg. Dry Extracts in Different Solvents"

_plants, 2021, doi:10.3390/plants10050909_

Round 1
Reviewer 1 Report
In this manuscript, the authors prepared extract from a lichen, Usnea barbata (L.) F.H. Wigg., using various solvents, and compared these characteristics. Acetone, ethyl acetate, ethanol, methanol, and water were used for solvent, and contents of usnic acid, total polyphenols, tannins in the extracts were measured. Antioxidant Activity of these extracts has a moderate correlation with usnic acid, tannins contents, and a low correlation with total polyphenols contents. In addition, lethality against brine shrimp (Artemia salina L.) larvae of these extracts was correlated with usnic acid contents.
The theme of this study is related in their previous studies, and there is no novel finding on the biological function of Usnea barbata (L.) F.H. Wigg. dry extracts and usnic acid. However, I think that some data shown in this study would be useful for the reader of this journal, Plants.
On the other hand, the data presentation and explanation of this manuscript do not seem to be reader friendly. These problems should be improved.
Specific points are listed below.
Comment #1: The manuscript is not properly divided into paragraphs. There are too many paragraphs in this manuscript, and at least, to divide into paragraphs after one line should be avoided.
Comment #2: This manuscript is not properly separated into subsections. It is better to combine the subsection 2.1 to 2.5, and indicate as "“Preparation of Usnea barbata (L.) F.H. Wigg. Dry Extracts and Determination of Contents of compounds” or something. In addition, it is better to show the table 2 to 4 as one table. The parameters, RT, Component Name, Area, Height, Weight of UBDE, Absorbance are not required for understanding the results of this study. To indicate contents of each compound per gram Usnea barbata (L.) F.H.Wigg. (starting material) in the table is helpful to understand this study.
Comment #3: The Material and Methods section should be described more concisely. Detailed protocols are not required to understand this study. Please cite proper references to concise the Materials and Methods section. At least, the subsections 4.1 to 4.4 should be described without separation into sub-subsections.
Comment #4: Table 7, the meaning of bold numeric values should be explained. Further, the usage of the abbreviations, C1-3 is not reader friendly. Please use "Fold dilution" or other expression way.
Comment #5: Figure 13 seems to be not required for understanding this study.
Comment #6: The order and the indication style of samples are not consistently indicated in the tables and the figures.
Comment #7: Figure 1-5 should be presented as one figure, with Figure 14 (the chromatogram chart loading standard solution of usnic acid). In addition, the bar marked with triangle, the line with x marks, and the numerical values indicated figures should be explained in the figure legend.
Comment #8: Line 71, the manufacturer's name “Sigma Aldrich” is not required in this line.
Comment #9: In the Conclusion section, the authors mentioned about antitumor properties of Usnea barbata (L.) F.H. Wigg. dry extract. However, antitumor effects of Usnea barbata (L.) F.H. Wigg. dry extract are not described in the Introduction section.
Comment #10: Antitumor activity of usnic acid has been demonstrated by many researchers. From this reason, in this manuscript, the authors seem to be focused on usnic acid. The authors should explain the reason why they focused on usnic acid in this study more clearly, and more detailed functional features of usnic acid should be explained in the Introduction section.
Comment #11: Line151, “Correlation between Metabolites Content and Antioxidant Activity” should be discarded. Please add proper explanations at the head of the next paragraph.
Comment #12: The trendlines for the correlation between total polyphenols content or tannins content and antioxidant activity are assumed to be changed by solvents used for study. Therefore, I am wondering the meaning to show the analysis results in this paper.
Comment #13: There is no conclusion which solvent is thought to be suitable for the extraction in this manuscript. Please discuss about the suitable extraction way for Usnea barbata (L.) F.H. Wigg. in the Discussion section.
Comment #14: There are many figures in this paper. However, some of these are only shown the same data in different way (e.g. Figure 8, 10, 11). The authors should reorganize figures. At least, Figure 8 and 10, or Figure 11 should be discarded.
Comment #15: The meaning to show Table 8 in this paper seems to be low. It is better to explain this table in sentences, such as "the mortalities were statistically significantly different between the each group".
Author Response
Please, see de attachment.

Reviewer 2 Report
Major comments: 1) Chemical analysis key parameters are missing. Hence, please supply validation data in the results section (you refer to quality control solution, which is this solution?). Since PDA was used peak purity index should be provided. In natural products the constituents are numerous and UV though important is not that selective. Hence, more data are needed to convince of the peak identified. Table 2 containing areas needs revision (it is not a proof of evidence of usnic acid identification). It is also critical to assess the matrix effect. Please extensively revise this part. It is a serious shortcoming of the article.
2. In the discussion, page 11, lines 270-271, please clarify the "phenolics" with non-polar character". It is a confusing comment.
Minor comments: The introduction begins abruptly and is too general. Please try to connect it more quickly with the main scope of the presented work. Please avoid continuous use of "we" and "could be". These repetitions hinder the reading of the manuscript.
Author Response
Please, see the attachment.

Reviewer 3 Report
Dear Authors,
The article Popovici et al. describes ‘Antioxidant and Cytotoxic Activities of Usnea barbata (L.) F.H. 2 Wigg. Dry Extracts in Different Solvents’. The manuscript presented by the authors is interesting and introduces some new elements. Particularly interesting is the cytotoxic effect of the tested plant material. Generally, however, proposes to extend the introduction to further present the research problem. The introduction mainly contains information about the activity of free radicals and the material under study. I miss presenting the innovation of the problem in the light of other studies. Generally, the article written good and solidly.
However, it contains a few of minor errors that must be corrected in order for the article to be published in Plants MDPI.
The following are the changes for improvement:
Abstract:
- In the case of the name Usnea barbata (L.) F.H.Wigg suggests to use the full name only the first time. in the rest of the text - only the abbreviation barbata. This is a general rule that is accepted in texts with Latin names.
- Expand the abbreviations DPPH in the abstract.
Key words:
1) Keywords should not be repeated in the title
Introduction:
- Line 61 - put a space: metabolites[15]
- Line 67 -73 – transfer explanation of abbreviations to abstract, e.g. total polyphenols (TPC), and tannins (TC). In introduction, only abbreviations can be used.
- Line 71 – „Sigma Aldrich” proszÄ™ podać tylko w metodyce.
- I propose to extend the introduction to information on barbata. Please emphasize the innovativeness of the research carried out. Are there any reports on the therapeutic effect and cytotoxicity of the tested plant material? Write more about the photochemistry of the plant material used.
Results:
- Lines 105-118 – the letters in the charts are illegible. I suggest reducing the X axis to 6 or 8 minutes.
- Lines 121 -125 – correct italics.
- I propose to standardize the number of decimal places in the tables. I suggest a third decimal place.
- Line 169 – IC50
Materiał i metody:
1) Line 361 – unreadable chart.
Why were such solvents used in the extraction?
After making all the changes, I recommend publishing in Plants MDPI.
Author Response
Please, see the attachment.
